# Investigation on the Stress and Deformation Evolution Laws of Shield Tunnelling through a Mining Tunnel Structure

**Entong Du** [1], **Lei Zhou** [1,*] and **Ruizhen Fei** [2]

1 MOE Key Laboratory of Deep Underground Science and Engineering, College of Architecture and Environment, Sichuan University, Chengdu 610065, China; duentong@outlook.com
2 China Railway Design Corporation, Tianjin 300142, China; ruizhenfei@yeah.net
* Correspondence: zhouleittkx@scu.edu.cn

**Abstract:** In the construction of a shield crossing an existing mined tunnel without load, it is imperative to develop corresponding design standards that reflect actual engineering force characteristics to ensure the successful completion of the tunnel construction. This study uses the MIDAS-GTS NX 2022 finite element software to facilitate the creation of a numerical model of a shield structure for an air-push-over mine tunnel project in Changsha, China while investigating the stress field's evolution during shield construction and calculating the maximum positive and negative bending moments and maximum axial forces for different structures and other force states under various construction conditions. This study's findings informed the design and construction optimisation of a shield tunnelling empty-push method. The outcomes of this numerical simulation led to several key findings: (1) The soil density exerted a significantly greater impact on the internal forces of the initial support structure than both the tunnel depth and soil Poisson's ratio. Additionally, a sudden shift in internal forces occurred within the 300–350 mm range when the lining thickness was altered. (2) Factors such as the tunnel depth, soil density, soil Poisson's ratio $\mu$, and lining thickness similarly influenced the internal forces of the segment and the initial support. Notably, the backfill layer thickness significantly affected the segment's maximum axial force, causing an abrupt change of approximately 300 mm. (3) It is essential to control the guide rail's thickness under the shield machine equipment's weight constraint to prevent it from becoming overly large.

**Keywords:** shield tunnelling; mining method; empty pushing; structural stress; deformation characteristics; numerical simulation

## 1. Introduction

Urban subway tunnels predominantly comprise shallow structures that traverse bustling city districts. The use of shield tunnelling methods during construction can effectively minimize the impact on ground surfaces. However, in the face of challenging geological conditions, such as hard rock, boulders, anchor cables, and varied layers of hardness and softness, shield tunnelling technology in China has not fully matured. As a result, employing shield tunnelling in these circumstances may expedite tool wear, diminish excavation speed, and necessitate more frequent tool replacements. This could lead to increased engineering costs and elevated safety risks. Considering these problems, it is prudent to consider a combined construction approach that integrates mining and shield tunnelling methods. Through the study of the combined construction method, optimization of the construction structure and engineering cost can be realized, and the safety hazards can be reduced, which is helpful in the successful completion of a tunnel project.

As the complexity of urban subway construction increases, new technical development demands have been raised for shield tunnelling construction. Research on the construction technology of shield tunnelling empty-pushing using the mining method has also made significant progress in China. In terms of construction technology research, as urban

subway construction becomes increasingly challenging, new technological development requirements have been proposed for the shield construction process. In China, shield tunnelling empty-pushing construction using the mining method has made significant progress. Regarding construction technology research, Duan et al. [1] determined the parameters of the Holmquist–Johnson–Cook model of the diorite in the Jinan area to ensure the accuracy of the rock-breaking simulation, using ANSYS to simulate the rock-breaking process to analyse the influence on the rock-breaking behaviour of the geometric configuration, including the blade width and blade fillet. Erharter et al. [2] studied the friction coefficient for tunnel-boring machine excavation planning by shearing rock specimens with different lithology to optimise the construction parameters of the shield in terms of friction. Choi et al. [3] experimentally evaluated the waterproofing performance of five sealant installation methods based on the type and number of layers of sealant, using a waterproofing performance test to determine the relative waterproofing performance of the tunnel tube sheets. Zhang et al. [4] investigated the mechanical properties of the new structural material FWP through compression and bending tests, determined the key design parameters of the bearing capacity and stiffness through numerical tests, and proposed a calculation method that can be used to calculate the bearing capacity and stiffness of FWP in practical engineering. Meanwhile, Suk min Kong et al. [5] compared the vibration measurement values generated when excavating the top surface using the existing NATM construction method and the TBM and NATM parallel construction methods, based on the vibration measurement data of the excavation site and using the NATM construction method; the vibration reduction effect of the two construction methods was analysed through 3D numerical analysis. Le et al. [6] proposed an equation describing the relationship between volume loss and the liquefaction potential index by monitoring field data obtained during the construction of the Binh Thanh-Su Tien tunnel on line 1 of the Ho Chi Minh City metro in Vietnam, which was used in practice as an indicator of potential large settlements caused by EBP tunnel boring machines in sandy soils. Jie et al. [7] examined the performance of blades made from 42CrMo low-alloy steel after different heat treatments, contributing valuable insights to fault prevention and cost reduction in shield machines. Numerous researchers domestically and internationally [8–10] have pioneered key construction technologies by modifying shield machine construction methods and overcoming the technical difficulties of shield machine construction under diverse geological conditions. Regarding numerical simulation and analysis, Imteyaz et al. [11] used the FEM-based software ABAQUS to analyse the deformation of specific rock mass characteristics under static and seismic conditions with and without lining. Islam et al. [12] used MIDAS GTS NX to carry out 3D finite element numerical simulations to optimize the geometric parameters and construction sequence of twin tunnels to help designers control the settlement caused during the excavation of back-loaded twin tunnels. Fang et al. [13] suggested a modelling method based on the coupled finite difference and discrete element methods, simulating the interaction between the shield machine and the layered rock mass. They further discussed the progressive failure mechanism of a layered rock mass. Moghtader et al. [14] developed an artificial neural network model that considered the non-linear relationship between the maximum surface settlement and 150 influential independent variables and collected real data from the Tehran Metro Line 16 project to build a training and test set to optimize the ANN technique, which eventually predicted the surface settlement of the above project accurately. Hussaine et al. [15] used the open-source AutoML framework to construct different machine-learning models to predict the maximum ground settlement when shield tunnels are constructed on soft subsoil, with advantages in terms of prediction accuracy. Xiao et al. [16] adopted four machine learning (ML) algorithms and four deep learning (DL) algorithms in shield machine attitude (SMA) prediction models, and finally integrated ML algorithms and DL algorithms to design a warning predictor for SMA according to the eight simulation results. Chen et al. [17] formulated a hybrid prediction dataset that incorporated geological and tectonic parameters. They based this on sampling methods using spatial and time series to obtain an approximate range of subsidence to reduce the

potential damage the project might inflict on the surrounding environment. Wang et al. applied the Matlab program of the BP neural network based on a genetic algorithm combined with engineering examples to achieve relatively efficient construction feedback through a forward analysis of construction parameters, which can effectively improve the response efficiency of unexpected conditions in the construction process. Many scholars [18–24] have used finite element numerical simulation, machine learning method comparison, and genetic algorithms to perform the numerical modelling of shield tunnelling under special external loads, investigating the deformation characteristics and mechanisms of tunnel structures. The results of the digital modelling are used to predict changes in construction conditions, optimize construction parameters, and control construction quality.

In summary, domestic research on shield machine empty-pushing through mining method in tunnel construction has primarily focused on aspects such as construction technology, quality control, and monitoring and measurement techniques. However, research contributions in structural stress and design are somewhat limited. In practical scenarios, as tunnelling projects progress, frequent stress disturbances occur within the structure. This results in substantial changes in the internal forces of the tunnel structure, which often fluctuate under varying building conditions. During the construction of a mine method tunnel project, the internal force fluctuations can exceed their limits, leading to potential failure or even collapse, of the existing structure. This poses risks to buildings surrounding metro lines and complicates tunnel construction. Ensuring safety through excessive building measures would inflate the project's cost. By studying the alternations in stresses within the tunnel structure in the current project situation, we can clarify the range and trend in potential stresses during construction. This facilitates the optimisation of the design criteria for the initial support and detailed structures such as tube sheeting, thereby achieving a balance between economic efficiency and safety. MIDAS-GTS software has a rich interface for importing solid files in various formats for creating realistic terrain and stratigraphic sub-interfaces by using a terrain data generator combined with borehole data, facilitating researchers' studies. At the same time, MIDAS-GTS can output combined envelope results such as vectors, section output clouds, and tables for analysis. Thus, in comparison to the methods used by Le et al. [6], Chen et al. [17], and other scholars, MIDAS-GTS have a better advantage in 3D modelling. Therefore, this study focuses on a section of Changsha Metro Line 3 in China. We use MIDAS-GTS finite element software to establish a model of tunnel construction using a shield machine with empty-pushing through mining method, and carried out a relevant mechanical analysis. Several numerical models were created following actual construction steps to examine the effects of different construction conditions on the displacements in the x, y, and z directions and the internal stress distribution within the shield-driven section. In comparison with other tunnelling projects in China, the simulation results of this study can offer recommendations for construction standards for shield machine empty-pushing through mining method in tunnel construction on the basis of construction safety. According to the simulation analysis results, we offer recommendations for construction standards for shield machine empty-pushing through mining method in tunnel construction. The research findings can maximise the load bearing capacity of the structure and ensure the economic efficiency of the project costs, which can provide some practical engineering data to support similar tunnelling projects in the future.

## 2. Materials and Methods

### 2.1. Project Overview

In a specific section of Changsha Metro Line 3 in China, the anchor cables utilised for the basement excavation support of a commercial plaza on the south side intersected with the shield tunnelling section, affecting an approximately 80 m stretch. Similarly, the anchor cables employed for the basement excavation support of an office building and shopping mall on the north side penetrated the shield tunnelling section, influencing a length of approximately 160 m. According to the geological data, field geological survey,

and drilling results from the area, the site is primarily covered by loose Quaternary strata, underlaid by chalk and muddy sand bedrock. The survey unveiled the presence of strongly and moderately weathered zones in gravel and fully weathered, strongly weathered, and moderately weathered zones in the muddy sandstone. The moderately weathered gravel exhibited a debris structure and a medium–thick-layered structure, whereas the muddy sandstone displayed a muddy sand structure, thick layered structure, and mudstone cementation. The geological distribution of the study sites is illustrated in Figure 1. The construction projects primarily involve vertical shaft construction, undermining horizontal tunnels, and undermining tunnels in the interval section, with the left and right boundaries of the undermining interval being left ZDK35 + 945.797, left ZDK36 + 117.911, right YDK35 + 951.006, and right YDK36 + 52.378, respectively. The geological distribution and construction plan for the undermined section of the station are depicted in Figure 1.

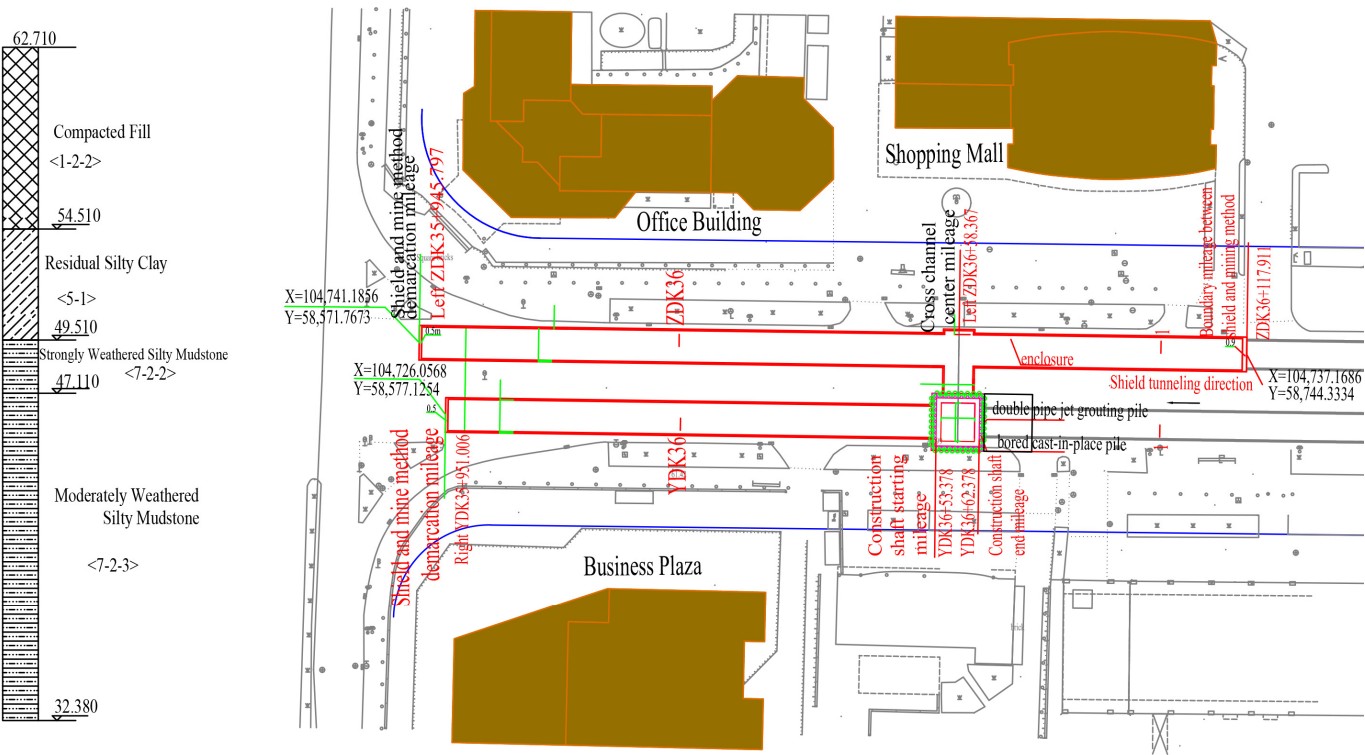

**Figure 1.** The engineering geological distribution map and plan view of the undermining section.

### 2.1.1. Preliminary Construction Scheme Comparison

Given that the basement pit support anchor cables encroach on the shield construction area on both sides of the concealed section, there is a risk of the shield cutter head becoming trapped. Initially, the use of manually excavated bored piles was proposed for removing anchor cables. However, owing to the smaller diameter of these piles, procuring or modifying the necessary anchor-cable extractor is challenging. Additionally, the poor linearity of anchor cables during pipe drilling might result in missed or severed cables [25]. Moreover, using manually excavated bored piles for anchor cable removal would occupy the main traffic routes, potentially causing traffic congestion. Therefore, during the construction drawing stage, this approach was optimized and adjusted. A combination of mining method excavation and shield tunnelling was chosen for anchor cable removal. This method ensures the complete removal of anchor cables within the intrusion section of the tunnel, thus facilitating the smooth passage of the shield.

### 2.1.2. Mining Method Tunnel Scheme

The initial support structure combined a steel mesh, system anchor bolts (advanced small conduits), gratings, and shotcrete for reinforcement. Upon completion, the initial

support construction for the mining method section, shield tunnelling empty-pushing, was conducted, with the gap between the tunnel lining segments and the initial support filled using grouting and gravel. A C35 concrete guide platform with a radius of 3200 mm and a thickness of 250 mm, was installed within a 60° range at the bottom of the cross section. To ensure the smooth passage of the tunnelling shield, the concrete guide platform should be accurately measured and inspected prior to the shield's advancement. Cross-sectional measurements of the tunnel excavation face should be taken at one-metre intervals, and any under-excavation should be promptly addressed. The final cross-sectional view of the mining tunnel is shown in Figure 2.

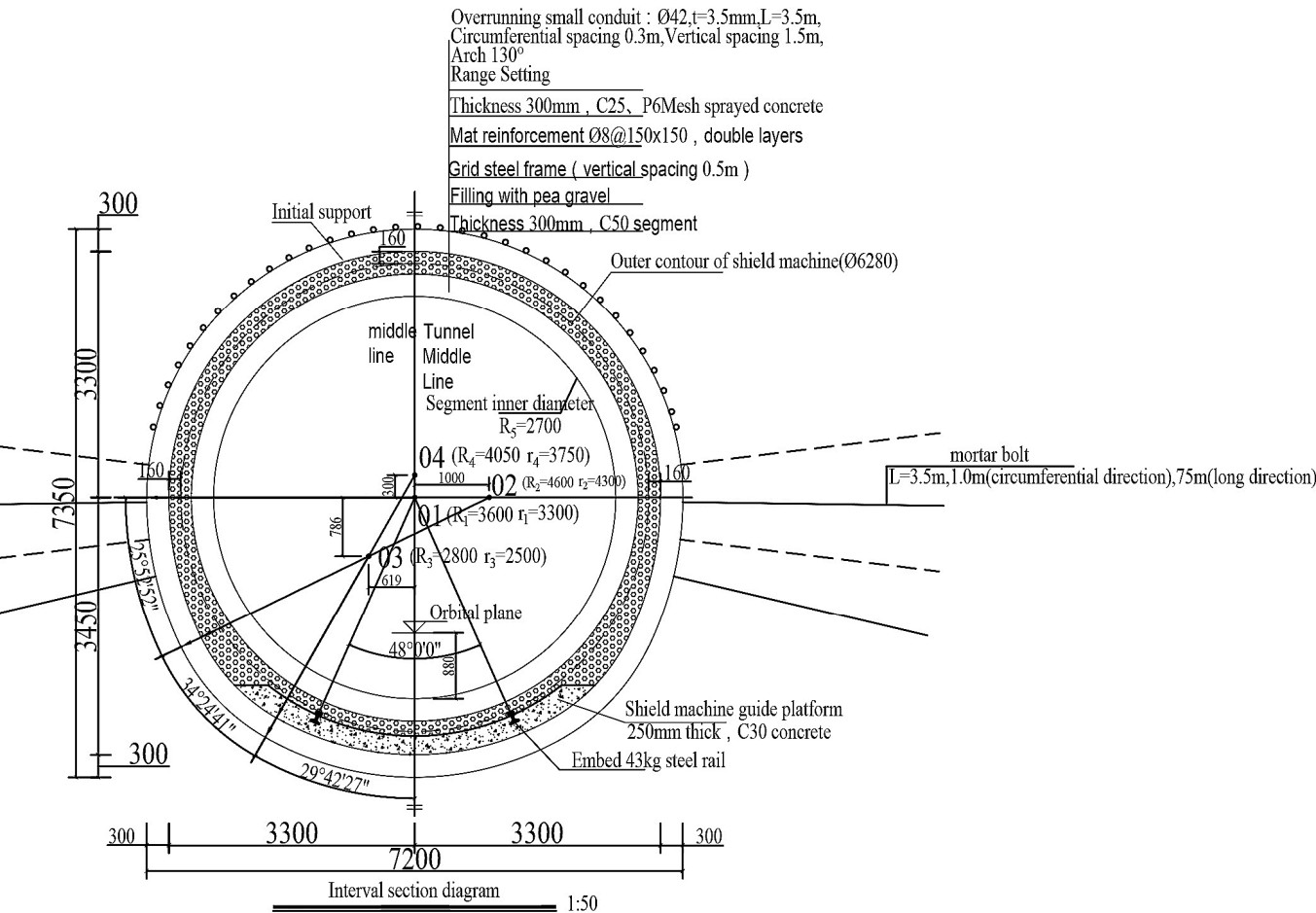

**Figure 2.** Cross-sectional view of the mining method tunnel.

## 2.2. Numerical Calculation Model and Method

A numerical model of the shield tunnelling empty-pushing through the mining method tunnel was established using the MIDAS-GTS finite element calculation software, considering the self-weight stress field for the ground stress field. During the modelling process, certain conditions were optimized and assumed, considering the following basic assumptions: According to the ISRM and other relevant Chinese standards, and concerning the advanced rock mechanics experiment method [26,27], the mechanical parameters of the soil layer at the construction site are measured to form a geological survey report. In the report, the corresponding physical and mechanical properties of representative soil layers are selected as the simulation parameters in the model, the detailed parameter settings are shown in Table 1. The entire construction process was simulated using the construction step sequence, considering the spatial displacement variation during construction, without considering the time effect.

**Table 1.** Physical and mechanical parameters of model materials.

| Soil and Rock Name | Natural Density $\rho$/(g/cm$^2$) | Cohesion c/kPa | Internal Friction Angle $\Phi$/(°) | Compression Coefficient /MPa$^{-1}$ | Deformation Modulus/MPa | Poisson's Ratio $\mu$ | Subgrade Reaction Coefficient | |
|---|---|---|---|---|---|---|---|---|
| | | | | | | | Vertical $K_v$/(MPa/m) | Horizontal $K_h$/(MPa/m) |
| Compacted Fill | 1.97 | 12.0 | 15.6 | 0.35 | / | / | 14 | 18 |
| Residual Silty Clay | 2.00 | 30.0 | 15.0 | 0.30 | 25 | 0.30 | 45 | 35 |
| Strongly Weathered Silty Mudstone | 2.29 | 45.0 | 28.0 | 0.20 | 120 | 0.25 | 150 | 140 |
| Moderately Weathered Silty Mudstone | 2.31 | 150.0 | 32.0 | / | 2500 * | 0.22 | 240 | 220 |

* Values with an asterisk (*) represent rock elastic modulus.

To minimize the effect of boundary constraints on the computational results, the boundary for each direction was established based on similar numerical simulations and experimental experiences for tunnels [28–31]. According to the principles of the numerical simulation, the boundary should be placed three to five times the diameter of the tunnel section outside the retaining structure. In this model, a distance four times the diameter of the tunnel section was adopted. The overall range of the numerical model was 70 m × 50 m × 40 m. The main stressed soil layers were simulated, from top to bottom, and their corresponding thicknesses were as follows: compacted fill (8.2 m), residual silty clay (5.2 m), strongly weathered silty mudstone (2.4 m), and moderately weathered silty mudstone (24.3 m extending from the bottom boundary of the model). The net distance between the left and right tunnel lines was 8.4 m. Each tunnel line had a horizontal width of 6.6 m and a vertical width of 6.75 m. For displacement boundary conditions, a free surface was used at the top of the model, and normal fixed constraints were employed for the other five faces in the vertical direction. Specifically, x and y normal fixed constraints were utilized for the horizontal constraints, and a z fixed constraint was applied to the bottom, as shown in Figure 3a. The modified Mohr–Coulomb criterion derived from Mahendra et al.'s [32] and Massinas et al.'s [33] closed solutions for the plastic zone and stress distribution around circular tunnels in the elastic-plastic semi-infinite plane based on the Mohr–Coulomb yield criterion and Li's [34] analysis of the Yumo railway project in China was adopted as the failure criterion for the grid elements using practical engineering experience as a basis. Solid elements were employed to simulate each soil layer. The initial support and shield segments were modelled using plate elements, whereas the gravel backfill layer and guide platform were simulated using solid elements. The numerical model corporates 8800 hexahedral mesh elements and 7601 nodes. Mesh size is controlled using a linear gradient, with a mesh size of 0.5 m for initial supports, segments, pea gravel, and soil inside the tunnel. Outside the tunnel, the mesh size for the soil gradually increases linearly with the distance from the tunnel centreline, reaching a maximum mesh size of 5 m.

To simulate the initial state of natural soil layers during the finite element modelling of foundation pit excavation, it is crucial to balance the initial stress. This implies that the soil model should only contain the initial stress field without any initial displacements. Initially, the soil displacement and stress fields under their own weights were initially calculated. Subsequently, to construct the initial stress field, the displacement zeroing function of MIDAS-GTS was utilized to remove the completed settlement displacement. Lastly, a finite element numerical model for each structure was established, as shown in Figure 3b. Ground displacement values for each geological condition simulated in the model were combined with stress–strain and modulus-of-elasticity formulas to derive the corresponding structural axial force and bending moment values.

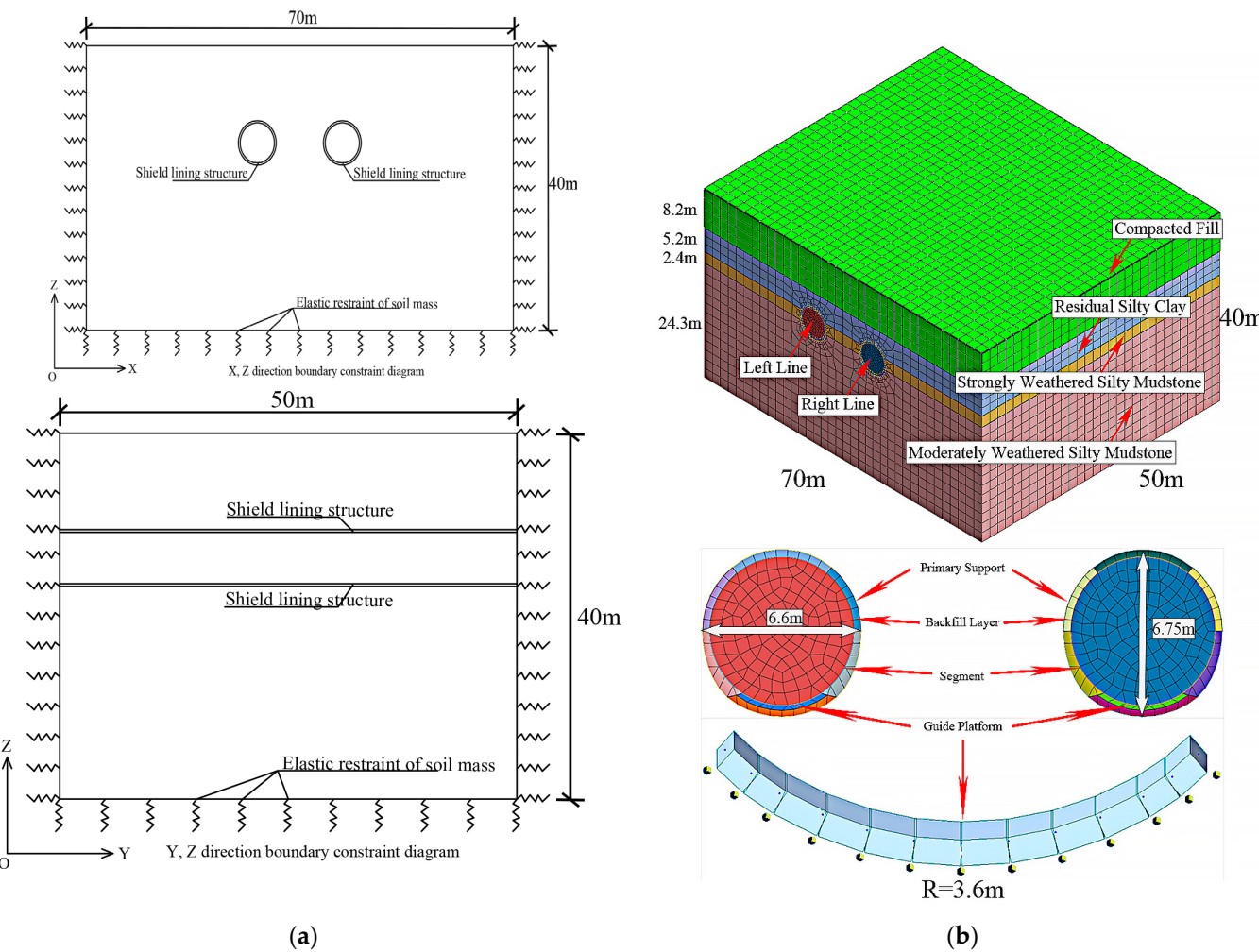

**Figure 3.** Finite element calculation models. (**a**) Finite element calculation models of each structure; (**b**) boundary conditions of finite element calculation models.

## 3. Results

### 3.1. Three-Dimensional Model Mesh Simulation Results

#### 3.1.1. Single-Tunnel Single-Ring Construction Process Numerical Simulation Results

During the numerical calculations, the specific construction steps for the tunnel model are as follows: Step 1: Achieving initial stress balance. Step 2: Excavating one ring in the left tunnel and implementing initial support and guide platform. Step 3: Backfilling gravel to the lower half of the initial support ring in the left tunnel. Step 4: Assembling one ring of segments in the left tunnel. Step 5: Completing the backfilling of gravel outside the segments. Step 6: Continuing excavation, support, guide platform construction, segment assembly, and gravel backfilling in the left tunnel until completion. Step 7: Continuing excavation, support, guide platform construction, segment assembly, and gravel backfilling in the right tunnel until completion. Taking the construction of the second section of the left tunnel as an example, the displacement cloud maps for each construction step are shown in Figure 4.

As can be seen from the above figure, the displacement changes in the x- and y-directions during the construction process of the left tunnel are very small, with the maximum displacement occurring at the horizontal and vertical axis positions ($x_{max}$ = 3.947 mm and $y_{max}$ = 3.60 mm), and the differences at the level of 0.01 mm. In the z-direction, the ground settlement gradually increases with the construction sequence, with the maximum displacement occurring directly above the tunnel, $z_{max}$ = 18.781 mm. The bulge at the bottom of the tunnel gradually decreases as the tunnel is enclosed and the overall stiff-

ness increases, but the magnitude of the change is around 0.01 mm, and the maximum displacement is 0.964 mm.

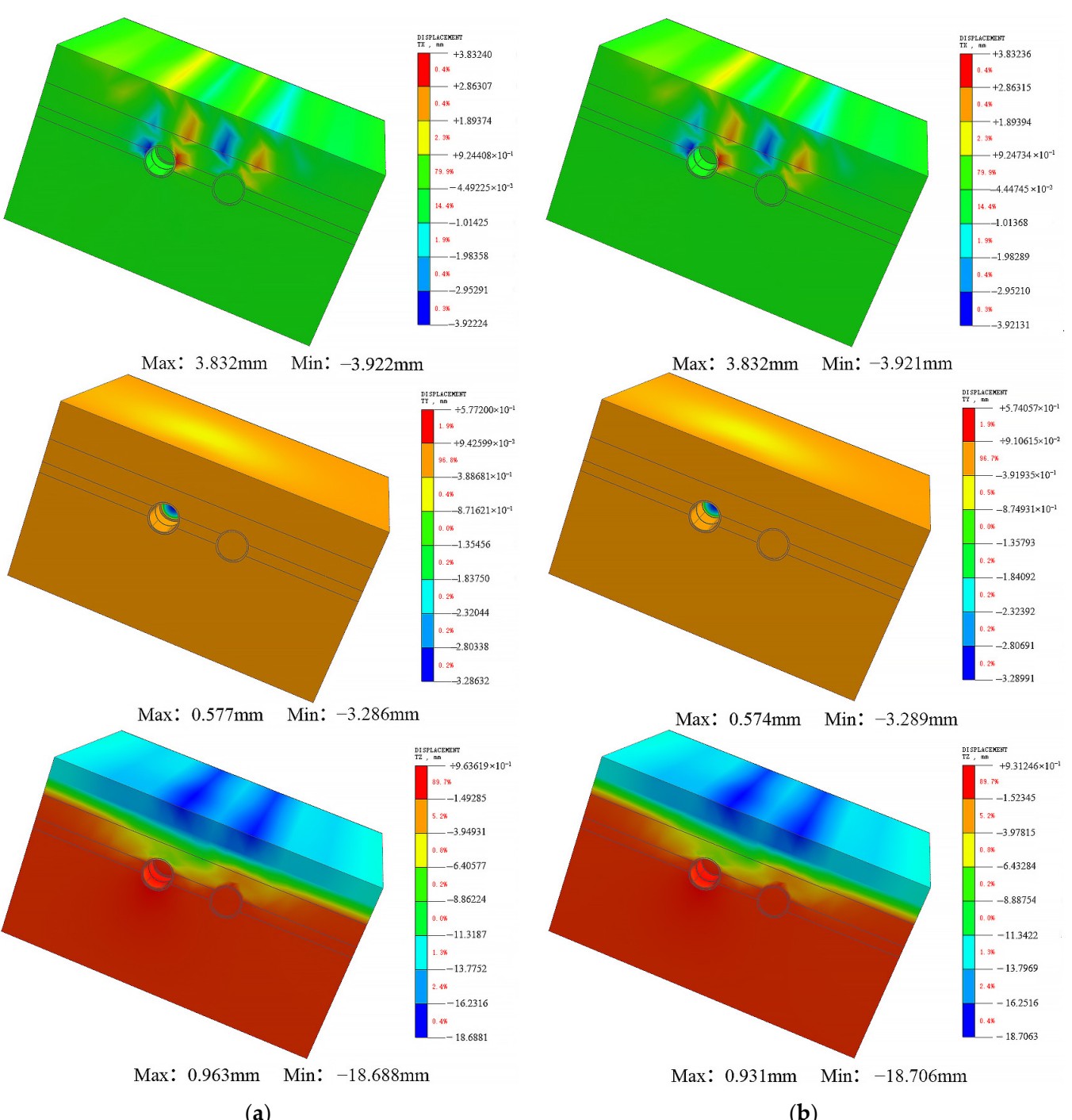

(**a**)                    (**b**)

**Figure 4.** *Cont.*

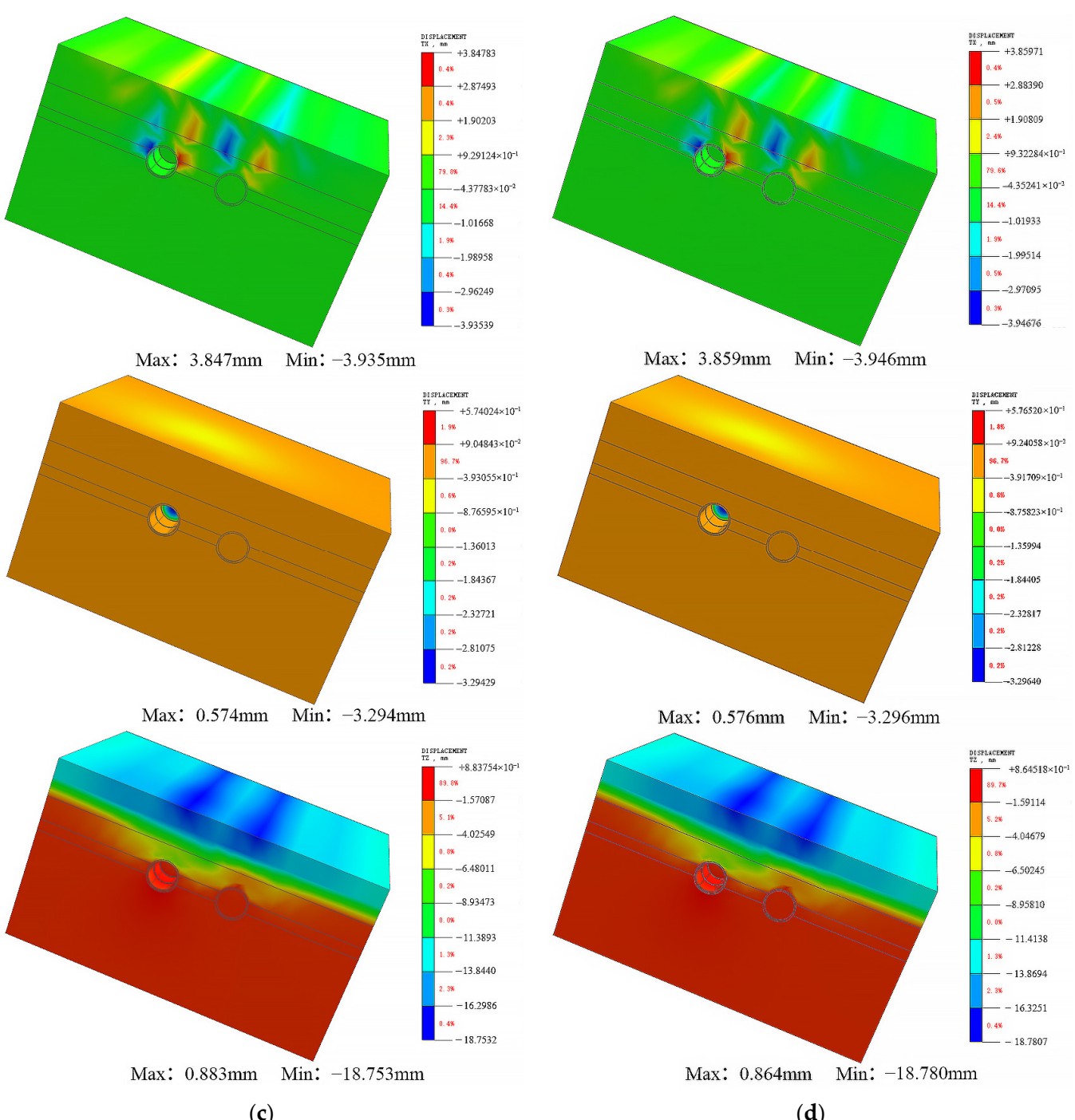

**Figure 4.** Displacement cloud map for the second construction stage. (**a**) the displacement values in the x, y and z directions after the initial support construction; (**b**) the displacement values in the x, y and z directions after the lower half of the gravel construction; (**c**) the displacement values in the x, y and z directions after the tunnel lining segment construction; (**d**) the displacement values in the x, y and z directions after the upper half of the gravel construction with a half symbol.

### 3.1.2. Single and Twin Tunnel Completed Numerical Simulation Results

In the construction step simulation, the y direction dimensions were 50 m, comprising a total of 10 excavation steps of 5 m each. Using the left tunnel as an example for the single tunnel part, the right line of the twin tunnel was simulated using the same construction steps as the left line. The final displacement cloud map for the completed single and twin tunnels is depicted in Figure 5.

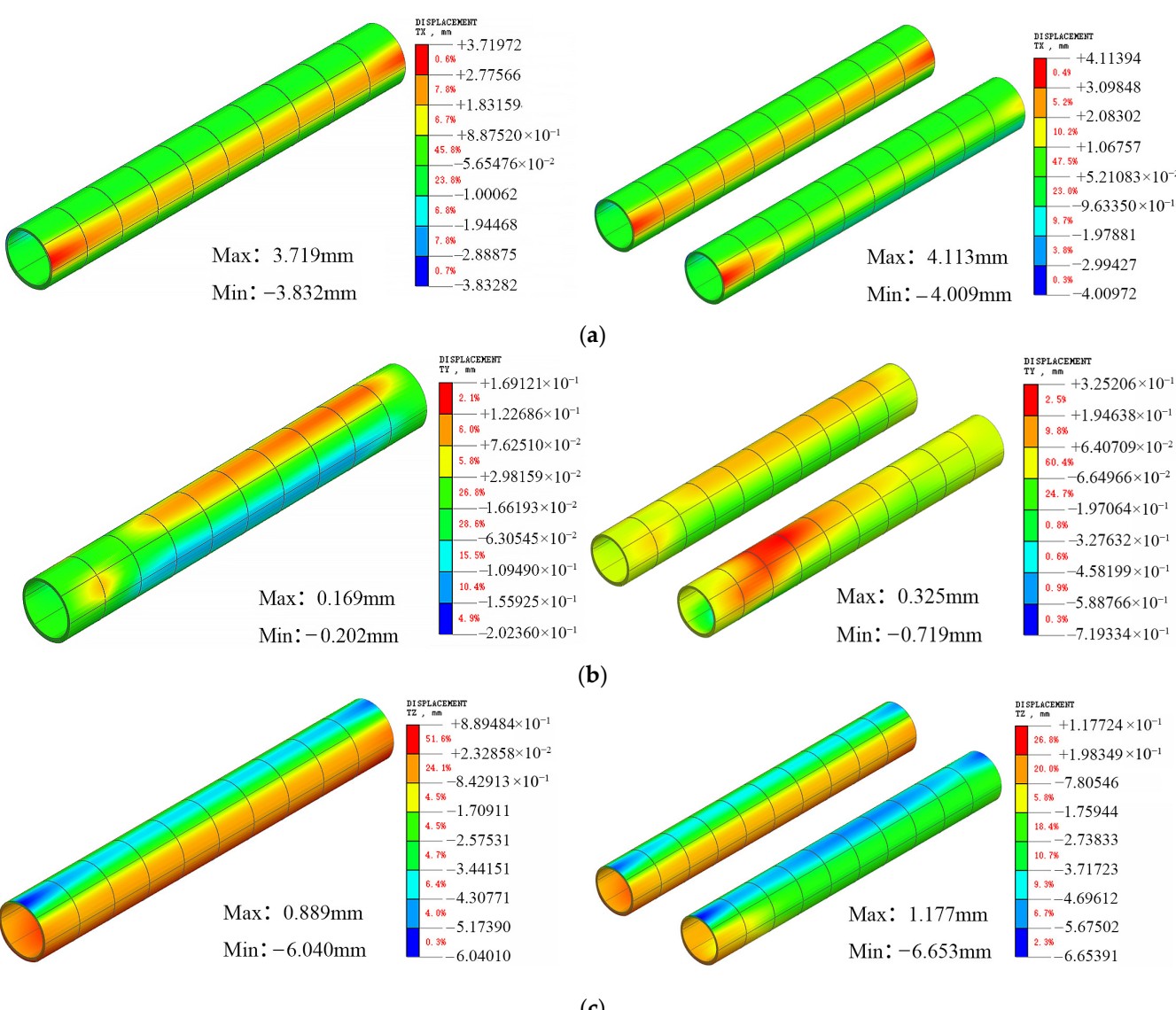

**Figure 5.** Tunnel x, y and z direction displacement values after single tunnel completion. (**a**) x direction displacement value of the tunnel after the completion of the left and right tunnel construction; (**b**) y direction displacement value of the tunnel after the completion of the left and right tunnel construction; (**c**) z direction displacement value of the tunnel after the completion of the left and right tunnel construction.

As can be seen from the above figure, following the completion of the left tunnel construction, the maximum displacement in the x-direction occurred at the horizontal axis position of the initial excavation, with $x_{max}$ = 3.834 mm. The maximum displacement in the y-direction occurred at the top and bottom of the tunnel in the vertical axis position. The maximum displacement occurred at the initial excavation position, with $y_{max}$ = 0.244 mm. The maximum displacement in the z-direction occurred at the top of the tunnel at the initial excavation position, with $z_{max}$ = 6.044 mm. All the aforementioned displacement values were within the allowable range. With the completion of the right tunnel construction, the maximum displacement in the x-direction occurred at the horizontal axis position of the initial excavation, with $x_{max}$ = 4.114 mm. The maximum displacement in the y-direction occurred at the top and bottom of the tunnel in the vertical axis position, with the maximum displacement occurring at the initial excavation position $y_{max}$ = 0.19 mm. The maximum displacement in the z-direction occurred at the top of the tunnel at the initial excavation position, with $z_{max}$ = 6.653 mm.

In this study, the limits of Schedule B.0.2 of the Technical Specification for Safety Protection of Urban Rail Transit Structures (CJJ/T202) [35] were adopted for the displacement control requirements of this model, as listed in Table 2. According to the simulation results, the displacement values are within the permissible range. Based on the simulation results presented above, it can be observed that during the construction of a single tunnel, the maximum displacement tends to occur at the top, bottom, and waist of the tunnel after completing the initial support, segmental lining installation, and gravel backfilling. The ground settlement gradually increases as the construction progresses, whereas the uplift at the bottom of the tunnel decreased after the tunnel was sealed by installing a segmental lining. Therefore, minimising the duration of the tunnel-sealing process is beneficial for controlling the uplift. Given that the net distance of the tunnel in this project is greater than one tunnel in diameter, after the single-hole and double-hole construction is completed, the location of the maximum displacement shifts; however, the relative location and absolute value of the displacement are closer to the results of the single tunnel.

**Table 2.** Urban rail transit structural safety control index values.

| Safety Control Indicators | Early Warning Value | Control Value |
|---|---|---|
| Tunnel Horizontal Displacement | <10 mm | <20 mm |
| Tunnel Vertical Displacement | <10 mm | <20 mm |

*3.2. Force Characteristics Results of Initial Support*

3.2.1. Impact of Tunnel Burial Depth on the Force Exerted on Initial Support

The Poisson's ratio is the ratio of the absolute value of the transverse positive strain to the axial positive strain when a material is subjected to unidirectional tension or compression. In geotechnics, Poisson's ratio is one of the parameters used to evaluate the properties of soils due to its sensitive response to the physical and mechanical state of the soil.

With the other simulation parameters held constant, the force on the initial support was calculated at different buried depths of H = 24, 26, 28, 30, 32, 34, and 36 m. The details of the force changes are illustrated in Table 3 and Figure 6a. The bending moment of the initial support was the highest at the top and bottom of the support ring and increased as the buried depth of the tunnel increased. The initial support section was under compressed, with the maximum axial force was located at the waist of the support ring. By controlling the variables, it was observed that the internal force of the initial support changed in the same manner, positively correlated with the buried depth of the tunnel. As the depth increased, it contributed to an increase in the maximum axial force.

**Table 3.** Initial support calculation results under different buried depths.

| H (m) | $M_{max}$ (kN·m) | $-M_{max}$ (kN·m) | $N_{max}$ (kN) |
|---|---|---|---|
| 24 | 131.35 | 138.74 | 854.12 |
| 26 | 146.05 | 154.30 | 951.33 |
| 28 | 149.36 | 158.61 | 981.30 |
| 30 | 154.97 | 165.93 | 1026.27 |
| 32 | 166.96 | 176.88 | 1102.92 |
| 34 | 171.87 | 180.66 | 1133.06 |
| 36 | 179.23 | 185.81 | 1158.42 |

3.2.2. Impact of Soil Bulk Density on the Force Exerted on the Initial Support

To study the influence of the soil unit weight ($\gamma$) on the initial support stress characteristics in the combined method, the soil unit weight was varied in the calculation model as 18, 19, 20, 21, 22, 23, and 24 kN/m. The resulting initial support stresses were obtained for different soil unit weights.

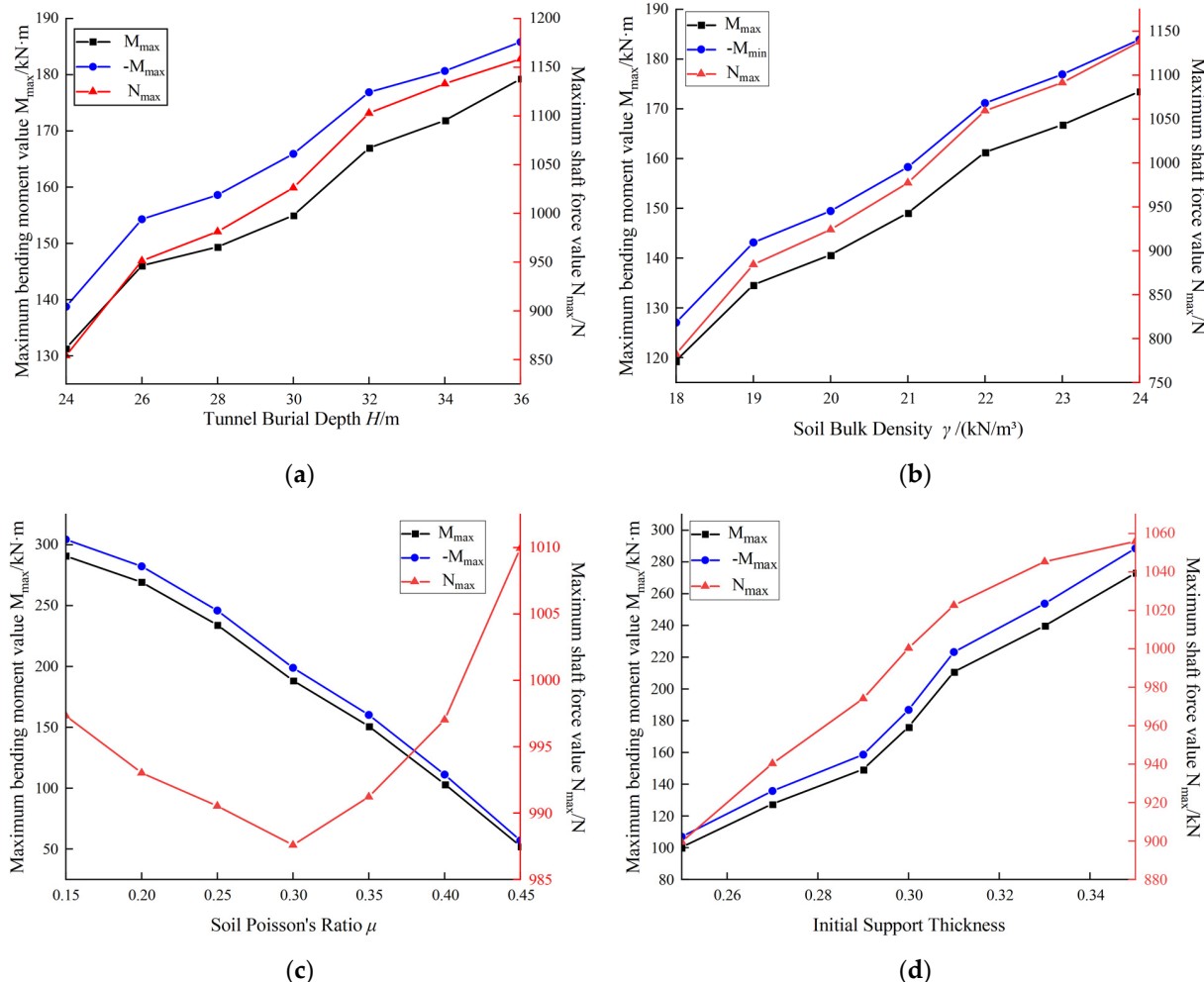

**Figure 6.** Initial support stress variation curve. (**a**) influence curve of tunnel burial depth on initial support stress; (**b**) influence curve of soil bulk density on initial support stress; (**c**) influence curve of soil Poisson's ratio on initial support stress; and (**d**) influence curve of initial support thickness on initial support stress.

As evident from the initial support stress variation curve in Table 4 and Figure 6b, both the maximum bending moment and axial force of the initial support section are linearly and positively correlated with the soil weight, with their values increasing as the weight increases. Moreover, as the unit weight parameter increases, the growth rate of the maximum axial force curve gradually outpaces that of the maximum bending moment curve. This suggests that, in this project, the influence of the unit weight on the axial force was more pronounced than that its influence on the bending moment.

### 3.2.3. The Influence of Soil Poisson's Ratio on the Forces Experienced by the Initial Support

The Poisson ratio of the soil is one of the parameters used to evaluate the performance of the soil. In this study, different Poisson's ratios were selected, with values of 0.15, 0.20, 0.25, 0.30, 0.35, 0.40, and 0.45, to investigate their influence on the initial support stress.

The analysis of the variation in internal force values shown in Table 5 and Figure 6c indicates that the maximum bending moment of the section was linearly and negatively related to the soil's Poisson's ratio, decreasing as Poisson's ratio increased. The variation in the maximum axial force value fluctuated less than that of the maximum bending moment value, initially decreasing and then increasing with an increase in Poisson's ratio. The minimum value was reached at a Poisson ratio of 0.3, indicating that the soil layer with a Poisson ratio of 0.3 had the least impact on the maximum axial force value of the initial support.

**Table 4.** Initial support calculation results under different soil bulk density.

| $\gamma$ (kN/m$^3$) | $M_{max}$ (kN·m) | -$M_{max}$ (kN·m) | $N_{max}$ (kN) |
|---|---|---|---|
| 18 | 119.38 | 127.04 | 782.54 |
| 19 | 134.61 | 143.14 | 884.30 |
| 20 | 140.66 | 149.46 | 924.05 |
| 21 | 149.08 | 158.30 | 977.33 |
| 22 | 161.26 | 171.15 | 1059.54 |
| 23 | 166.78 | 176.91 | 1091.48 |
| 24 | 173.45 | 183.91 | 1137.92 |

**Table 5.** Initial support calculation results under different soil poisson's ratio.

| $\mu$ | $M_{max}$ (kN·m) | -$M_{max}$ (kN·m) | $N_{max}$ (kN) |
|---|---|---|---|
| 0.15 | 290.76 | 304.28 | 997.32 |
| 0.20 | 269.23 | 282.26 | 993.02 |
| 0.25 | 234.05 | 245.98 | 990.53 |
| 0.30 | 188.46 | 198.85 | 987.59 |
| 0.35 | 150.86 | 160.21 | 991.22 |
| 0.40 | 103.41 | 111.18 | 997.02 |
| 0.45 | 52.47 | 57.09 | 1009.94 |

3.2.4. The Influence of the Initial Support Thickness on the Forces Experienced by the Initial Support

In the mining method, the structural stiffness of the initial support ring was directly affected by the thickness of the initial support. The internal forces of the initial support at various thicknesses were obtained in the simulation by adjusting the thickness of the initial support in the tunnel model to 0.25, 0.27, 0.29, 0.30, 0.31, 0.33, and 0.35 m. The maximum bending moment and maximum axial force of the initial support are positively correlated with thickness and increase with thickness, as shown in Table 6 and Figure 6d. However, there is a discrepancy in the slope of the bending moment and axial force curves as the thickness increases. This indicates that the maximum bending moment of the starting section is influenced by its thickness; the maximum bending moment changes more significantly than the maximum axial force; and the maximum bending moment changes abruptly.

**Table 6.** Initial support calculation results under different initial support thickness.

| Initial Support Thickness (m) | $M_{max}$ (kN·m) | -$M_{max}$ (kN·m) | $N_{max}$ (kN) |
|---|---|---|---|
| 0.25 | 100.33 | 106.89 | 899.61 |
| 0.27 | 127.63 | 135.73 | 940.32 |
| 0.29 | 149.36 | 158.61 | 974.12 |
| 0.3 | 176.14 | 186.81 | 1000.38 |
| 0.31 | 210.74 | 223.25 | 1022.65 |
| 0.33 | 239.76 | 253.73 | 1045.32 |
| 0.35 | 272.91 | 288.53 | 1055.90 |

*3.3. Force Characteristics Results of Segment*

3.3.1. The Influence of the Tunnel Burial Depth on the Segment Forces

The burial depth of the tunnel not only impacts the initial support but also influences the segment through the lining. By modifying the burial depth H of the tunnel to 24, 26, 28, 30, 32, 34, and 36 m while keeping the other simulation parameters of the soil layer constant, the evolution of the bending moment and axial force in the segment was obtained, as illustrated in Table 7 and Figure 7a. From the analysis, it can be observed that the variation trend of the internal force variation in the segment is similar to that of the initial support when the burial depth of the tunnel changes. However, compared to the initial

support, the increase in the tunnel's burial depth had a more substantial effect on the growth trend of the internal forces in the segment. This indicates that the range of burial depth should be appropriate in this project to balance the fluctuation effects on the segment and the initial support.

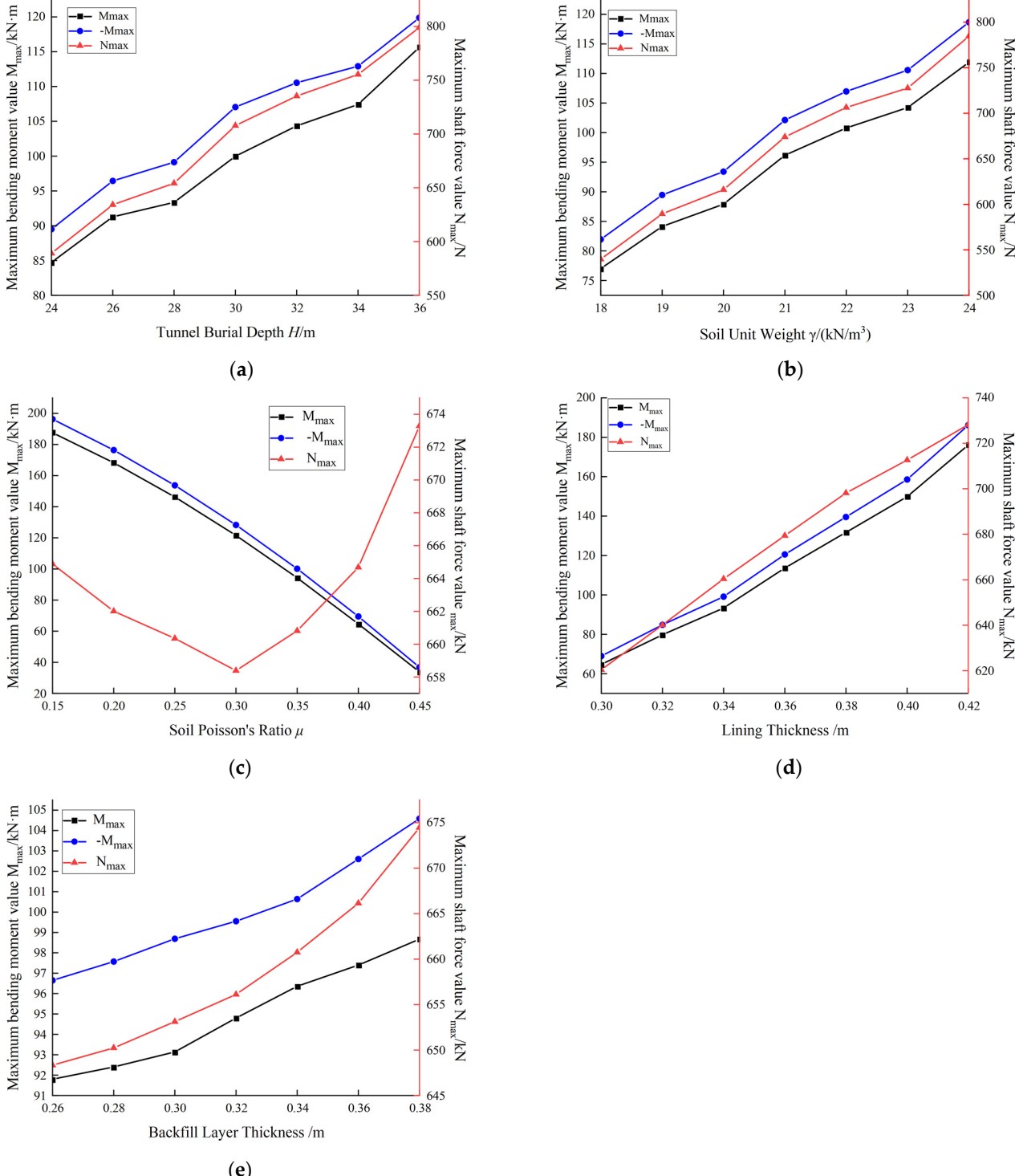

**Figure 7.** Segment stress variation curve. (**a**) Influence curve of tunnel depth on segment stress; (**b**) influence curve of soil unit weight on segment stress; (**c**) influence curve of soil Poisson's ratio on segment stress; (**d**) influence curve of lining thickness on segment stress; and (**e**) influence curve of backfill thickness on segment stress.

**Table 7.** Segment calculation results under different buried depths.

| H (m) | $M_{max}$ (kN·m) | $-M_{max}$ (kN·m) | $N_{max}$ (kN) |
|---|---|---|---|
| 24 | 84.74 | 89.51 | 589.05 |
| 26 | 91.28 | 96.44 | 634.22 |
| 28 | 93.35 | 99.13 | 654.20 |
| 30 | 99.98 | 107.05 | 707.77 |
| 32 | 104.35 | 110.55 | 735.28 |
| 34 | 107.42 | 112.91 | 755.37 |
| 36 | 115.63 | 119.88 | 798.91 |

3.3.2. Influence of Soil Unit Weight on Segment Forces

To further study the effect of soil bulk density (γ) on the stress characteristics of the segment using the combined method, the soil bulk density in the calculation model was altered to 18, 19, 20, 21, 22, 23, and 24 kN/m. The trend in stress variation in the segment under different soil bulk densities was obtained, as illustrated in Table 8 and Figure 7b. The maximum bending moment and axial force in the segment showed a positive correlation with the soil bulk density, and their values increased with increasing density. This indicates that the impact of soil bulk density on the internal forces in the segment was relatively balanced in this project.

**Table 8.** Segment calculation results under different soil bulk density.

| γ (kN/m³) | $M_{max}$ (kN·m) | $-M_{max}$ (kN·m) | $N_{max}$ (kN) |
|---|---|---|---|
| 18 | 77.02 | 81.96 | 539.68 |
| 19 | 84.13 | 89.46 | 589.53 |
| 20 | 87.91 | 93.41 | 616.03 |
| 21 | 96.18 | 102.13 | 674.02 |
| 22 | 100.79 | 106.97 | 706.36 |
| 23 | 104.24 | 110.57 | 727.65 |
| 24 | 111.90 | 118.65 | 784.77 |

3.3.3. Influence of Soil Poisson's Ratio on Segment Forces

To investigate the impact of the soil's Poisson's ratio (μ) on the force characteristics of the segments, we selected soil Poisson's ratios of 0.15, 0.20, 0.25, 0.30, 0.35, 0.40, and 0.45. We then studied the effect of different Poisson ratios on the stress of the segment. The variation in the maximum bending moment and axial force in the section with changes in the soil's Poisson's ratio is illustrated in Table 9 and Figure 7c. It can be observed that the maximum bending moment is negatively linearly correlated with the soil's Poisson's ratio, with its value decreasing as the Poisson's ratio increases. In comparison, the variation in the maximum axial force value is relatively stable. It first decreases and then increases with an increase in Poisson's ratio, reaching a minimum value at a Poisson's ratio of 0.3. This pattern is consistent with the variation observed in the initial support. In terms of soil poison, according to the analysis of this special trend in change, when the Poisson's ratio is less than 0.3, because the vertical strain is greater than the horizontal strain, the soil has less constraint on the horizontal displacement of the excavated structure. The excavation structure is an arch structure, and the members should be mainly compressed. The vertical displacement and horizontal displacement of the structure increase with the increase in depth, and the axial force tends to decrease. When the Poisson's ratio reaches 0.3, the vertical deformation growth rate of the soil around the excavation structure decreases, while the lateral deformation gradually converges and becomes smaller when the lateral earth pressure gradually increases, so the axial force increases. But in general, the change in Poisson's ratio has little effect.

**Table 9.** Segment calculation results under different soil poisson's ratio.

| μ | $M_{max}$ (kN·m) | $-M_{max}$ (kN·m) | $N_{max}$ (kN) |
|---|---|---|---|
| 0.15 | 187.59 | 196.31 | 664.88 |
| 0.20 | 168.27 | 176.41 | 662.01 |
| 0.25 | 146.28 | 153.74 | 660.35 |
| 0.30 | 121.59 | 128.29 | 658.39 |
| 0.35 | 94.29 | 100.13 | 660.81 |
| 0.40 | 64.63 | 69.49 | 664.68 |
| 0.45 | 33.85 | 36.83 | 673.29 |

3.3.4. Influence of Lining Thickness on Segment Forces

Because the thickness of the lining was mainly determined by the thickness of the segments, the thickness of the segments also affected their stress situations. By varying the thicknesses of the segments in the tunnel model to 0.30, 0.32, 0.34, 0.36, 0.38, 0.40 and 0.42 m, the internal force changes in the segment under different thicknesses were obtained and are shown in Table 10 and Figure 7d. When the thickness of the segment was different, the distribution of the internal forces in the segment was the same. The maximum bending moment and axial force of the segment were both positively correlated with the thickness of the segment and increased with increasing thickness. Moreover, the change in the maximum axial force was more in line with the linear relationship than the maximum bending moment, indicating that the change in the axial force was more stable under the influence of the segment thickness.

**Table 10.** Segment calculation results under different lining thickness.

| Lining Thickness (m) | $M_{max}$ (kN·m) | $-M_{max}$ (kN·m) | $N_{max}$ (kN) |
|---|---|---|---|
| 0.30 | 64.73 | 68.96 | 620.42 |
| 0.32 | 79.77 | 84.83 | 640.10 |
| 0.34 | 93.35 | 99.13 | 660.40 |
| 0.36 | 113.64 | 120.52 | 679.40 |
| 0.38 | 131.71 | 139.53 | 698.14 |
| 0.40 | 149.85 | 158.58 | 712.65 |
| 0.42 | 176.07 | 186.15 | 728.21 |

3.3.5. Influence of Backfill Layer Thickness on Segment Forces

To provide structural strength, backfill materials, such as gravel and cement slurry, must be used to fill the space between the segments and the initial support while building a shield tunnel that passes through a mining region. As shown in Table 11 and Figure 7e, the stress curves of the segments with various backfill layer thicknesses were produced to analyse the impact of the backfill layer on the stress of the segments. The backfill layer thicknesses were 0.26, 0.28, 0.30, 0.32, 0.34, 0.36 and 0.38 m. It can be observed that the maximum bending moment of the segment has a positive correlation with the backfill layer thickness and increases as the layer thickness increases. Although the total size of the change is minimal, the change in the thickness of the backfill layer has a significant impact on the rate at which the maximum axial force increases.

*3.4. Stress Behaviour of Guide Platform*

3.4.1. Influence of Shield Body Weight on Guide Platform Stress

The weight of the shield per metre has an impact on the load applied on the guide platform, which consequently affects the force exerted on the guide platform. In order to analyse this, the stress of the segments at different burial depths was calculated, by varying the weight of the shield per metre to 350, 400, 450, 500, and 550 kN/m. The results,

displayed in Table 12 and Figure 8a, demonstrate that the internal forces acting on the guide platform were altered. The maximum bending moment of the guide platform exhibited a linear and positive correlation with the weight of the shield, with its value increasing as weight increased. Since the weight has a greater influence on the maximum positive bending moment compared to the maximum negative bending moment of the guide platform, the disparity between the maximum positive and negative bending moments increases with the shield weight. Consequently, this leads to an uneven distribution of the internal bending moments within the guide platform.

**Table 11.** Segment calculation results under different backfill layer thickness.

| Backfill Layer Thickness (m) | $M_{max}$ (kN·m) | $-M_{max}$ (kN·m) | $N_{max}$ (kN) |
| --- | --- | --- | --- |
| 0.26 | 91.80 | 96.65 | 648.34 |
| 0.28 | 92.40 | 97.57 | 650.24 |
| 0.30 | 93.14 | 98.69 | 653.12 |
| 0.32 | 94.80 | 99.55 | 656.12 |
| 0.34 | 96.36 | 100.64 | 660.75 |
| 0.36 | 97.40 | 102.60 | 666.14 |
| 0.38 | 98.67 | 104.58 | 674.46 |

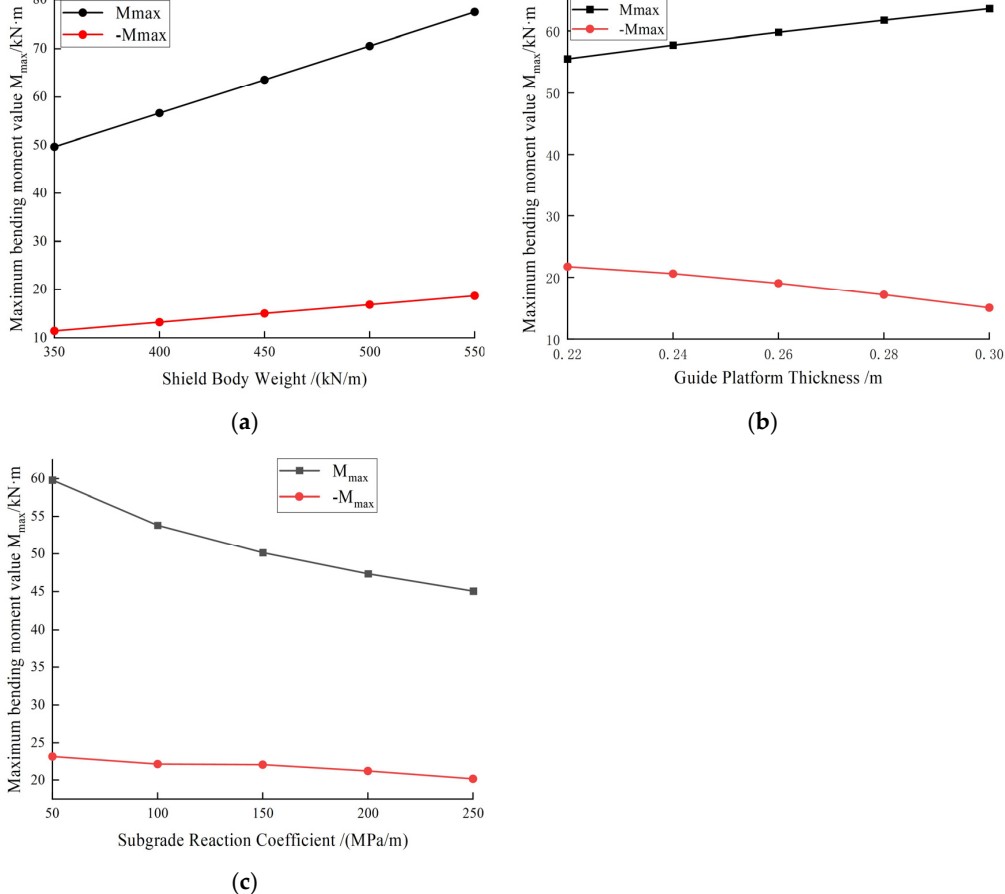

**Figure 8.** Guide platform stress variation curve. (**a**) Influence curve of shield body weight on guide platform stress; (**b**) influence curve of guide thickness on guide platform stress; and (**c**) influence curve of subgrade reaction coefficient on guide stress.

**Table 12.** Guide platform calculation results under different shield body weight.

| Shield Body Weight (kN/m) | $M_{max}$ (kN·m) | $-M_{max}$ (kN·m) |
|---|---|---|
| 350 | 49.68 | 11.43 |
| 400 | 56.64 | 13.23 |
| 450 | 63.60 | 15.04 |
| 500 | 70.56 | 16.84 |
| 550 | 77.52 | 18.65 |

3.4.2. Influence of Guide Platform Thickness on Guide Platform Stress

The thickness of the guide platform directly impacts its stiffness; thus, the thickness of the guide platform in the tunnel model was altered to 0.22, 0.24, 0.26, 0.28, and 0.30 m. The internal force variation curves of the guide platform with different thicknesses are depicted in Table 13 and Figure 8b. It is evident that the maximum positive bending moment of the guide platform exhibits a linear relationship with the thickness of the guide platform, with its value increasing as the thickness increases. Conversely, the maximum negative bending moment of the guide platform demonstrates a linear negative correlation with the thickness of the guide platform, with its value decreasing as the thickness increases. As the weight of the shield body has an opposing effect on the maximum positive and negative bending moments, the difference between them increases. Consequently, the uneven distribution of the maximum positive and negative bending moments within the guide platform becomes more pronounced.

**Table 13.** Guide platform calculation results under different guide platform thickness.

| Guide Platform Thickness (m) | $M_{max}$ (kN·m) | $-M_{max}$ (kN·m) |
|---|---|---|
| 0.22 | 55.53 | 21.75 |
| 0.24 | 57.74 | 20.61 |
| 0.26 | 59.80 | 19.04 |
| 0.28 | 61.75 | 17.15 |
| 0.30 | 63.60 | 15.04 |

3.4.3. Influence of Subgrade Reaction Coefficient on Guide Platform Stress

The bed coefficient represents the pressure required to induce a unit deformation per unit area in a geotechnical body under an external force. It plays a role in determining the settlement degree of a structure to some extent. In order to study the impact of different soil bed coefficients on the force acting on the guide platform, values of 50, 100, 150, 200 and 250 MPa/m were selected. The resulting change curve of the internal forces on the guide platform is presented in Table 14 and Figure 8c. Analysis reveals that the maximum positive and negative bending moments of the guide platform exhibit a linear and negative relationship with the bed coefficient. An increase in the bed coefficient has a suppressing effect on the bending moments experience by the guide platform. Notably, the influence of the bed coefficient on the maximum positive bending moments is more pronounced in this project. Consequently, it leads to a gradual equalisation of the maximum positive and negative bending moments within the guide platform.

**Table 14.** Guide platform calculation results under different subgrade reaction coefficient.

| Subgrade Reaction Coefficient (MPa/m) | $M_{max}$ (kN·m) | $-M_{max}$ (kN·m) |
|---|---|---|
| 50 | 59.80 | 23.08 |
| 100 | 53.82 | 22.09 |
| 150 | 50.14 | 22.01 |
| 200 | 47.35 | 21.18 |
| 250 | 45.07 | 20.15 |

## 4. Discussion

This study aims to optimize the stress characteristics of a real project through a numerical model and investigate the stress variation pattern of certain structures in the air-thrust section of the shield. The findings provide suggestions for the structural design criteria in the shield air-thrust over-mine method tunnel project. However, it should be noted that this study only considers some selected engineering geological profiles as references for the model, and the material properties of the structures have been optimized, neglecting the properties of specific substrates or material interactions. Incorporating the numerical simulation results into the actual project and further improving the structural design criteria will be the main focus of future engineering construction.

Considering that the structural criteria for shield empty thrust sections can vary based on different project construction conditions, the recommendations provided in this study for partial structural criteria during construction are relevant but not universally representative. Namely, the results of this study are applicable to the specific conditions of the current project. From the above analysis and discussion, it is evident that by selecting appropriate structural criteria corresponding to the stress variation intervals and considering the impact of each engineering geological condition on the tunnelling project, the range of stress variation can be effectively controlled. This allows for the optimal utilization of the load-bearing capacity of each structure while ensuring structural integrity. By combining practical project construction experience and employing a well-founded structural design, it is often possible to achieve cost savings in engineering. This can be achieved through the effective control of the initial support size, the implementation of suitable pipe sheets and other necessary structures, a reduction in construction material input, and the selection of more appropriate processing techniques. These measures contribute to the economical construction of engineering projects.

In order to help future work, the following suggestions are put forward according to the research results.

### 4.1. Recommendations for Initial Support Design Criteria

In contrast to conventional practices that rely on increasing burial depth or reducing overburden thickness to address engineering safety and economic concerns, a more rational determination of burial depth for interval tunnels can be achieved by integrating strata distribution with software analysis. Furthermore, the simulation analysis reveals a positive correlation between the bending moment and axial force of the initial support with the burial depth of the tunnel, displaying a similar pattern of change. However, the growth rate of internal forces decreases after reaching a certain depth. Considering the geological conditions specific to this project, the impact of burial depth on the internal forces of the initial support structure is concentrated in the range of 33–35%, whereas the impact of soil bulk density is concentrated in the range of 44–45%. This indicates that, within a fixed soil layer, the bulk density factor has a greater influence on the internal forces of the initial support structure compared to the burial depth factor. Therefore, when determining the tunnel path, it is crucial to control the burial depth within the range of approximately 22 kN/m$^3$, where bulk density undergoes significant changes. This approach ensures the optimal utilization of the structure's load-bearing capacity under substantial burial depths while considering the impact of soil bulk density on the initial support structure in this project.

Moreover, the soil Poisson's ratio $\mu$ has a distinct impact on axial force changes, differing from other factors. In this project, the range of $\mu$ variation falls within 0.25–0.3, which corresponds to the declining section of the internal forces in the initial support. Thus, it has minimal detrimental effects on this structure. Regarding the structural dimensions, it is notable that the lining thickness experiences a sudden change in the internal forces of the initial support within the range of 300–350 mm. Therefore, a thickness of 300 mm can be selected for the initial support in this project, allowing for the full utilization of the structure's load-bearing capacity while ensuring an economical choice of materials.

*4.2. Recommendations for Segments Design Criteria*

The simulation analysis revealed similar effects of tunnel depth, soil density, soil Poisson's ratio μ, and lining thickness on the internal forces of both the segments and the initial support. The impact of burial depth on the internal forces of the initial support structure was concentrated in the range of 33–36%, while the influence of soil density was concentrated in the range of 44–45%. The soil Poisson's ratio μ also affected the internal forces of the initial support, as it fell within the declining stage. On the other hand, the effect of lining thickness on the segment's forces was weaker compared to the initial support.

Overall, the changes in conditions within this project had a lesser impact on the internal forces of the segments compared to the initial support. Therefore, the design standards for the segments can complement those of the initial support. Although the influence of backfill thickness on the internal forces of the segments was relatively balanced and concentrated in the range of 4–7%, it is crucial to control the backfill void to be no less than approximately 300 mm. This is the point where the maximum axial force mutation occurs, and maintaining this threshold helps prevent structural stress mutations. It is evident that the time factor plays a crucial role in the assembly of pipe pieces into rings and gap backfill processes. This factor significantly impacts the forces and displacements experienced by the pipe pieces. Therefore, further investigation into the use of integrated assembled structural bodies in similar projects is recommended for future stages.

*4.3. Recommendations for Guiding Platform Design Criteria*

The purpose of setting the guide rail is as follows: (1) Control the advance direction of the shield machine. (2) Transform the weight of the shield body into the uniform load that the primary support can bear, so as not to cause excessive deformation or local cracking damage in the primary support during the shield advance process, resulting in engineering accidents. If too-large guide rails are used, there will be greater requirements for the size of the shield machine, which will lead to an increase in the amount of backfill materials required. While increasing the project investment, it is not conducive to the rapid closure of the structure, which will eventually lead to a decrease in safety. It can be seen that it is of great significance to study the determination of guide parameters.

Through the analysis of the influence of the weight of the shield and the thickness of the guide platform on the internal forces of the guide platform, it was observed that the variations in internal forces caused by changes in the shield's weight were amplified by a factor of 1.54. Similarly, the variations in internal forces resulting from changes in the guide platform's thickness were amplified by a factor of 1.43. Therefore, considering the limitation of the shield machine equipment's weight, it is advisable to avoid excessively large thicknesses for the guide platform. For this project, a recommended control range for the guide platform's thickness would be around 200–250 mm.

Additionally, taking into account the planned tunnel burial depth and the range of the soil bed coefficient in this project (150–250 MPa/m), it can be observed that the project is situated in a stage where there is a balance between the positive and negative maximum bending moments. This is a favourable condition for the construction of this project.

**5. Conclusions**

To address the challenges arising from the construction of a shield pushing through a mining method tunnel, this study introduces a novel approach by conducting numerical calculations and analysis using finite element methods. It focuses on evaluating the combined structural system comprising the initial support, shield tube sheet, and shield guide table. According to the study, some suggestions are put forward for tunnel construction, and the conclusions are as follows:

(1) The bulk density of the initial support in the soil layer has a greater influence on its internal force than the buried depth, while Poisson's ratio has a weaker influence on the structure. The influence of the former should be considered first in the construction; that is, the excavation should be carried out in the soil with a depth of about 22 kN/m$^3$,

and the initial support's thickness of 300 mm should be selected. Compared with the other initial support parameters of the tunnel [36,37], material savings and structure optimization are realized, and the mechanical performance of the initial support is fully exerted.

(2) The variation law of the segment under the influence of various factors is similar to that of the initial support and is weaker than that of the initial support. Therefore, the segment design standard can assist in the formulation of the initial support design standard. The research shows that the backfill gap of the segment should be controlled to be 300 mm corresponding to the mutation point, which is similar to the parameter law in the existing research [38], so as to prevent damage to the structure caused by the sudden change in internal force.

(3) The weight of the shield machine is often limited by equipment specifications. Therefore, the control points of the design standard are mainly placed in the thickness of the guide platform, and the specific thickness is 200–250 mm. Different from the analysis of the thrust system [39], length [40], and other parameters of the shield machine by most scholars, this study provides a reference for the design standard of guide platforms.

**Author Contributions:** Conceptualization, E.D. and L.Z.; methodology, R.F.; software, R.F.; validation, E.D., L.Z. and R.F.; formal analysis, R.F.; investigation, E.D.; resources, R.F.; data curation, R.F.; writing—original draft preparation, E.D.; writing—review and editing, E.D. and L.Z.; visualization, E.D.; supervision, L.Z. All authors have read and agreed to the published version of the manuscript.

**Funding:** This work was financially supported by the National Natural Science Foundation of China (52204104); the Sichuan Science and Technology Program (2023YFH0022); the Opening Fund of State Key Laboratory of Geohazard Prevention and Geoenvironment Protection, Chengdu University of Technology (SKLGP2021K009); the Major research and development project of Metallurgical Corporation of China LTD in the non-steel field (2021-05); the Sichuan University postdoctoral interdisciplinary Innovation Fund.

**Institutional Review Board Statement:** Not applicable.

**Informed Consent Statement:** Not applicable.

**Data Availability Statement:** The datasets generated for this study are available from the corresponding author upon reasonable request.

**Acknowledgments:** The authors of this work are grateful for the experimental help provided by the Key Laboratory of Deep Earth Science and Engineering, Ministry of Education.

**Conflicts of Interest:** The authors declare no conflict of interest.

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
