# Peer review of "Investigation on the Stress and Deformation Evolution Laws of Shield Tunnelling through a Mining Tunnel Structure"

_applsci, doi:10.3390/app13148489_

Round 1

Reviewer 1 Report (Previous Reviewer 1)

The manuscript has good quality to be considered publishing in this Journal. 

Author Response

Reviewer 2 Report (Previous Reviewer 2)

Begin by clearly stating the purpose of the study and its relevance to the field of tunnel construction. Explain why developing design standards based on actual engineering force characteristics is important and how it can contribute to the successful completion of tunnel projects. Briefly describe the specific characteristics and challenges of the tunnel project in Changsha, China, that served as the basis for the numerical model. This will help readers understand the practical implications of the study. Provide more details about the numerical modeling approach using MIDAS-GTS software. Explain how the model was constructed, including the selection of parameters and assumptions made. This will enhance the transparency and reproducibility of the study. Restructure the paragraph discussing the findings to clearly present each key finding as a separate point. Start each point with a clear statement summarizing the finding, followed by relevant details and supporting evidence. This will make the findings easier to understand and follow. After presenting the key findings, discuss their practical implications for the design and construction optimization of shield tunnelling using the empty-push method. Explain how the findings can inform decision-making processes and potentially improve the efficiency, safety, or cost-effectiveness of similar tunnel projects. Consider expanding the discussion by providing further analysis or insights based on the findings. This could involve comparing the results to existing design standards, discussing potential limitations or challenges, or exploring future research directions. Elaborate on the relationship between the weight constraint of the shield machine equipment and the required thickness of the guide rail. Explain the potential consequences of an overly large guide rail and why it is important to control its thickness.
See the attached file for other comments

Proofread to correct English grammar and spellings

Author Response

Reviewer 3 Report (Previous Reviewer 3)

The authors did termendensouly efforts to improve the MS quality, However, few more references for literature review may provide the article in excellent shape which are following- Stability Analysis of Shallow Depth Tunnel in Weak Rock Mass: 3D Numerical Modeling Approach. Authors should mentioned what types of international standards used to estimate the Physical and mechanical properties of rock mass, i.e, ASTM or ISRM or any Chines standards. Also authors may include few references for laboratory testing of rock specimen which may be- Digital rock physics and application of high-resolution micro-CT techniques for geomaterials; Correlation of ultrasound velocity with physico-mechanical properties of Jodhpur sandstone.

Author Response

This manuscript is a resubmission of an earlier submission. The following is a list of the peer review reports and author responses from that submission.

Round 1

Reviewer 1 Report

Du et al., (Study on Partial Structural Stress Characteristics and Design Standards of Tunnel Structure by Shield Tunneling Empty Pushing through Mining Method) propose the study of stress field evolution of the shield tunnelling empty pushing through numerical simulations in commercial software. After reading the full manuscript and comparing it with the finding reported in the literature, it is concluded that the current form of the summited work has no novelty. The simulation was achieved in commercial software, and the proposed modified Mohr-Coulomb model is already reported in the literature. Furthermore, there was no scientific discussion. Because Applied Science is a recognized journal with an impact factor of 2.8, the summited manuscript lacks the merit to be published in this Journal.

On the other hand, there are several English mistakes in the manuscript. Figures have very low resolution, and the results are poorly structured and discussed. Current trends in this kind of simulation can be found in doi.org/10.1016/j.tust.2022.104852; doi.org/10.1016/j.autcon.2022.104732,doi.org/10.1155/2020/6836492 (this include data validation)

there are several English mistakes in the manuscript

Reviewer 2 Report

The manuscript discusses a study on the stress field evolution of the shield tunneling empty pushing through undermining method in subway construction, using numerical analysis. The study found that factors such as soil density, tunnel depth, soil Poisson's ratio, lining thickness, and backfill layer thickness have an impact on the internal forces of the initial support and the segment. Additionally, the study suggests that the thickness of the guide rail should be controlled to optimize the design and construction of the shield tunneling empty pushing. Overall, the document is well-structured and provides a clear overview of the study's objectives and findings. It would be helpful to include more context or explanations for readers unfamiliar with the terminology and concepts used in the study. Additionally, the implications and potential applications of the study's findings could be highlighted to make the document more informative and engaging for a wider audience.

Here are some suggestions for improving the document:

1.       Clarify the purpose of the study: While the document briefly mentions that the study aimed to investigate the stress field evolution of the shield tunneling empty pushing through undermining method, it would be helpful to provide more context on why this is important. For example, the document could explain how understanding the stress field evolution can help optimize the design and construction of shield tunnels, or how it can improve safety and efficiency during construction.

2.       Simplify technical language: The document uses technical language that may be difficult for non-experts to understand. To make the document more accessible, you could try to explain technical terms or use simpler language where possible. For example, instead of saying "soil Poisson's ratio," you could say "how compact the soil is."

3.       Provide more explanation of numerical analysis: The document mentions that numerical analysis was used to simulate excavation construction, but it is unclear what this means or how it was done. Including an explanation of numerical modeling setup, boundary conditions, mesh selection, constitutive model, rock mass properties, analysis, and how it was used in this study would help readers understand the methodology.

4.       Highlight implications of findings: The document briefly mentions that the study's findings were used to optimize the design and construction of the shield tunneling empty pushing, but it does not explain how or why. Adding a sentence or two on the implications of the findings would help readers understand why the study is important and how it can be applied in practice.

5.       Use headings or bullet points: Breaking up the document into headings or bullet points can make it easier for readers to follow the main points and understand the structure of the study. This can also make it easier to skim the document for important information.

6.       Some figures are not very clear, the text is not readable.

7.       Too many figures also figure caption numbering is not correct.

8.       Relate your results and discussion with existing literature

9.       Add conclusions of this study

10.   Add recommendations and future work

11.   Review the latest papers on the subject and add them

Reviewer 3 Report

The authors tried to investigate the stress field evolution of the shield tunneling empty pushing through undermining method segment structure using numerical analysis based on a shield tunneling empty pushing section of a subway. FEM tools used for numerical modelling of Changsha Metro Line 3. Various properties measured and work as input in numerical modelling which were tunnel depth, soil density, soil Poisson's ratio μ, and lining thickness. Based on the numerical calculation results, the design and construction of the shield tunneling empty pushing were optimized. According to the geological data and field geological survey and drilling results of the area, the site is primarily covered by loose Quaternary strata, with underlying bedrock consisting of chalk and muddy sand. Authors described the importance and necessity of tunnels in bustling city with good amount of references. However, few more references for literature review may provide the article in excellent shape which are following- Stability Analysis of Shallow Depth Tunnel in Weak Rock Mass: 3D Numerical Modeling Approach.

Authors should provide figure 1 clear details in English.

Table 1 Physical and mechanical parameters of model materials- authors should be mentioned what types of international standards are used to estimate the Physical and mechanical properties of rock mass, i.e, ASTM or ISRM or any Chines standards. Also, authors may include few references for laboratory testing of rock specimens which may be- Digital rock physics and application of high-resolution micro-CT techniques for geomaterials; Correlation of ultrasound velocity with physico-mechanical properties of Jodhpur sandstone.

Authors should mention which types of failure criteria are used for FEM numerical modeling and also may cite some references like- Finite Element Analysis of Road Cut Slopes using Hoek & Brown Failure Criterion; Designing Cut Out Distance for Continuous Miners Operation using Numerical Modelling and Rock Mechanics Instrumentation.

Line 190- Step 3: Backfilling gravel at the 3 o'clock and 9 o'clock positions in the left tunnel.. What is the timing meaning here? What is the relation with modeling?

Line 239- All of the above displacement values are within the allowable range. How to verify it?

Round 2

Reviewer 1 Report

Du et al., (Study on Partial Structural Stress Characteristics and Design Standards of Tunnel Structure by Shield Tunneling Empty Pushing through Mining Method) propose the study of stress field evolution of the shield tunnelling empty pushing through numerical simulations in commercial software.

Comment 1

Abstract need to be modified. Only add the more relevant information.

Comment 2.

The introduction section is too basic. Consider recent publications about finite element simulation, machine learning, and genetic algorithms. Moreover, it would be best to highlight the benefit of your work with those reported in the literature.

Comment 3

The author missed important information about simulation: Figure 3 presents a mesh, but the quality is not the best (this needs to be improved),  boundary conditions, assumptions, and solution setup.

Comment 4

The document does not include any type of data validation of the obtained results.

Comment 5

Please, discuss in detail what will be the benefit of your work in comparison with the algorithms or methods discussed in doi.org/10.1016/j.tust.2022.104852; doi.org/10.1016/j.autcon.2022.104732; doi.org/10.1155/2020/6836492.
